# A Deep Predictive Coding Network for Learning Latent Representations

## Abstract

It has been argued that the brain is a prediction machine that continuously learns how to make better predictions about the stimuli received from the external environment. It builds a model of the world around us and uses this model to infer the external stimulus. Predictive coding has been proposed as a mechanism through which the brain might be able to build such a model of the external environment. However, it is not clear how predictive coding can be used to build deep neural network models of the brain while complying with the architectural constraints imposed by the brain. In this paper, we describe an algorithm to build a deep generative model using predictive coding that can be used to infer latent representations about the stimuli received from external environment. Specifically, we used predictive coding to train a deep neural network on real-world images in a unsupervised learning paradigm. To understand the capacity of the network with regards to modeling the external environment, we studied the latent representations generated by the model on images of objects that are never presented to the model during training. Despite the novel features of these objects the model is able to infer the latent representations for them. Furthermore, the reconstructions of the original images obtained from these latent representations preserve the important details of these objects.

## 1 Introduction

The general idea of predictive coding (Mumford, 1991; 1992; Pennartz, 2015) postulates that the brain is continuously trying to predict the information it receives from external environment. An implementation of predictive coding was first proposed as a model of visual information processing in the brain (Rao & Ballard, 1999). Recently, it was described as an implementation of the free-energy principle in the brain (Friston, 2008). Predictive coding models the visual information processing pathways as a recurrently connected hierarchical neural network. Feedback connections from higher to lower level areas convey predictions about the activities of the lower level neurons and feedforward connections convey the residual errors in these predictions to higher level areas.

Several studies have focused on the biological plausibility of predictive coding and its relation to other learning approaches. In Spratling (2008), the author showed that a model of biased competition (Desimone & Duncan, 1995) that uses lateral inhibition to suppress the input of other nodes is equivalent to the linear model of predictive coding. An extension to predictive coding has been proposed in Spratling (2012) that relaxes the requirement of symmetric weights between two adjacent layers in the network. In a similar study, it was shown that the error-backpropagation and predictive coding use similar forms of weight changes during learning Whittington & Bogacz (2017).

From the perspective of training deep neural networks, predictive coding is an approach that is widely supported by neurophysiological data (Jehee & Ballard, 2009) and adheres to the architectural and locality (in terms of learning) constraints imposed by the brain. Existing studies on predictive coding has focused on small neural network models to study the development of orientation selective receptive fields in primary visual cortex (Rao & Ballard, 1999; Spratling, 2012). It is unclear how predictive coding can be used to build deep neural network models of the brain to study more complicated brain processes like attention, memory, etc. Another important question that arises while building models of the brain is how can we comply with the architectural constraints applicable in the brain like the retinotopic arrangement of receptive fields that is found in

the sensory cortical areas. At present, mostly neural networks with fully connected layers are used, which implies that the receptive fields of neurons are as big as the field of view. To overcome this, neural network models are trained on patches from real world images. This approach works well when training small neural network models but it is difficult to extend it for training deep neural networks.

In this paper, we present a systematic approach for training deep neural networks using predictive coding in a biologically plausible manner. The network is used to learn hierarchical latent representations for a given input stimulus. The architecture of these neural networks is inspired by convolutional neural networks (LeCun et al., 1998). However, to comply with the retinotopic arrangement of receptive fields observed in sensory cortical areas, we employ neural networks in which filters are *not* applied across the entire layer, similar to locally connected layers used in Taigman et al. (2014). Instead, filters are applied only to a small receptive field which allows us to train the filters associated with different receptive fields independently. This approach can be easily scaled to train deep neural networks for modeling information processing along the sensory processing pathways.

In general, the approach proposed in this paper can be used for stimuli in any modality. To illustrate the effectiveness of the approach, we trained a deep neural network using predictive coding on 1000 real-world images of horses and ships from the CIFAR-10 data set. The model is trained in an unsupervised learning paradigm to build a generative model for real-world images and is used to infer latent representations for real-world. To estimate the capacity of the network in modeling real-world images, we used the model to infer latent representations for new images of horses and ships as well as objects that are never presented to the network during training. The model is able to reconstruct the original real-world images from the inferred latent representations while retaining the important features of the objects in these images. This shows that the model can capture the causal regularities in real-world images.

The paper is organized as follows: Section 2 describes the architecture and the predictive coding based learning algorithm used for training deep neural network models. Section 3 describes the results of studies conducted using the trained models. Section 4 discusses the computational implications of deep predictive coding and its relationship with other approaches in machine learning. Section 5 summarizes the conclusions from the experiments reported in this paper.

## 2 MODEL

Suppose, we have a set of training images $(\mathbf{x}_1, \cdots, \mathbf{x}_i, \cdots)$ where $\mathbf{x}_i \in R^{W \times H \times C}$. The aim of the learning algorithm is to learn a generative model that can be used to infer the latent representations for the training images and other images that have not been used in training.

### 2.1 ARCHITECTURE

Consider a neural network with $(N + 1)$ layers where 0 represents the input layer and $N$ represents the topmost layer in the network. The input layer is used to present the training images to the network. Figure 1 shows a section of this network that depicts the connections between the layer $l$ and the layers above $(l + 1)$ and below $(l - 1)$ it. The neurons in a given layer $(l)$ are arranged in a 3-dimensional block of shape $Y_l \times X_l \times K_l$. Here, $Y_l$, $X_l$ and $K_l$ denote the height, width and the number of channels in layer $l$, respectively. The neurons in the layers $l$ and $(l + 1)$ are connected through $K_{l+1}$ filters of size $D_l$ and a stride of $s_l$. Based on this, the height and width of the layer $(l + 1)$ are given as

$$Y_{l+1} = \frac{(Y_l - D_l)}{s_l} + 1 \qquad (1)$$

$$X_{l+1} = \frac{(X_l - D_l)}{s_l} + 1 \qquad (2)$$

The number of channels in layer $(l + 1)$ is equal to the number of filters between the layers $l$ and $(l + 1)$.

The architecture of the network in Figure 1 bears some resemblance to to the architecture of a Convolutional Neural Networks (CNNs). However, there are two important differences between CNNs and the neural network used in this paper:

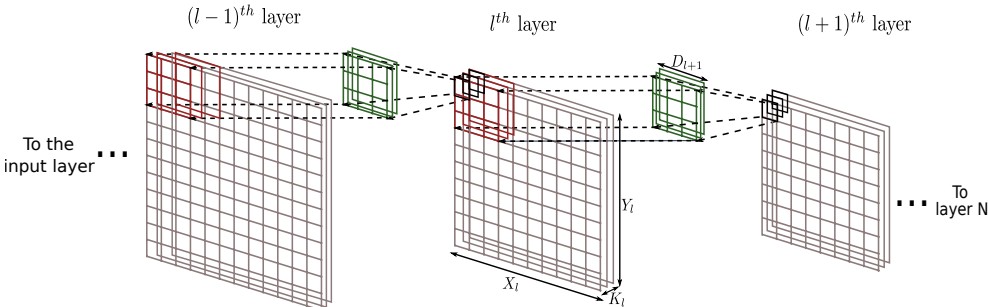

Figure 1: Architecture of the deep predictive coding neural network

- The neurons in a given layer in the network, shown in Figure 1 project to neurons only in their receptive field. This implies that the neurons in a particular channel in layer $l$ are connected to the neurons in layer $(l + 1)$ through filters that are learned independently.

- The most important difference is in the direction of information propagation with respect to CNNs. In a conventional CNN, the information propagates from layer 0 to layer $N$ and during learning the error gradients propagate from layer $N$ to layer 0. In contrast, in our predictive coding network the information propagates from layer $N$ to layer 0 in the network shown in Figure 1 and the error gradients propagate in the opposite direction.

To better understand the structure of connections between layer $l$ and the layer $(l - 1)$, let us denote the output of the neurons in the $m^{th}$ row and the $n^{th}$ column (here, referred to as $(m, n)$) of layer $l$ as $y_{m,n}^{(l)}$ which is a vector with $K_l$ elements. Based on this, the output of the neurons in layer $(l - 1)$ is given as

$$\hat{y}_{(s_{l-1}m+i),(s_{l-1}n+j)}^{(l-1)} = \phi(\mathbf{w}_{m,n,i,j}^{(l)} y_{m,n}^{(l)}), \quad \begin{array}{l} i, j \in \{1, \cdots, D_{(l-1)}\}, \\ m \in \{1, \cdots, Y_l\}, n \in \{1, \cdots, X_l\} \end{array} \tag{3}$$

where $\mathbf{w}_{m,n,i,j}^{(l)}$ denotes the filters through which the neurons at position $(m, n)$ in layer $l$ project to the position $(s_{l-1}m + i, s_{l-1}n + j)$ in layer $(l - 1)$. The filter $\mathbf{w}_{m,n,i,j}^{(l)}$ will be a matrix with dimensions $K_{l-1} \times K_l$. $\phi$ represents a non-linear vector-valued activation function with $K_{(l-1)}$ elements.

It may be notes that when the stride is less than the filter size, it results in an architecture with overlapping receptive fields. As a result, neurons in layer $l$ project to the overlapping positions in layer $(l - 1)$. Therefor, to determine the output of neurons in layer $(l - 1)$ we compute the average of the projections made by the layer $l$. This procedure is analogous to unpooling in a deconvolution network (Zeiler et al., 2010) in order to retain the dimensions of a layer.

## 2.2 Learning Algorithm

In this paper, the classical methodology of predictive coding Rao & Ballard (1999) is employed to train a generative neural network model that can be used to infer the latent representations of a given input image. For a given input image ($\mathbf{x}_i$), the latent representations at layer ($l$) in the network are learned such that they can accurately predict (using Equation 3) the latent representations at the layer below ($l - 1$). The learned representations at layer $l$ serve as target for learning the latent representations at layer ($l + 1$) in the network.

Suppose $\mathbf{y}_l$ and $\hat{\mathbf{y}}_l$ represent the actual and predicted latent representations for the neurons in layer $l$ of the network, then the total error ($E$) for all the layers in the network is given as

$$E = \sum_{l=0}^{N} \left( \ell_p(\mathbf{y}^{(l)} - \hat{\mathbf{y}}^{(l)}) + \ell_p(\mathbf{y}^{(l)}) + \sum_{m,n,i,j} \ell_p(\mathbf{w}_{m,n,i,j}^{(l)}) \right) \tag{4}$$

where $\ell_p(.)$ denotes the loss computed in accordance with $p$-norm. The total error in Equation 4 includes both the loss due to prediction and the regularization loss. Note that the limits of the

summation in Equation 4 are from $0$ to $N$ (instead of $0$ to $(N + 1)$). This is because there is no layer that learns to predict the activities of the neurons in the topmost layer of the network.

The total error in Equation 4 is used to simultaneously learn the latent representations and the synaptic weights in the model such that the prediction error at each layer in the network is minimized. This implies that the latent representations at a particular layer in the network try to capture the information present in the latent representations at the layer below. This allows us to train a deep generative model of the external stimulus presented to the network. To explicitly include the aspect of the network architecture with non-shared weights in the total error, the error in Equation 4 is expanded as:

$$E = \sum_{l=0}^{N} \left( \sum_{m,n}^{Y_l, X_l} \ell_p(y_{m,n}^{(l)} - \hat{y}_{m,n}^{(l)}) + \sum_{m,n}^{Y_l, X_l} \ell_p(y_{m,n}^{(l)}) + \sum_{m,n,i,j} \ell_p(\mathbf{w}_{m,n,i,j}^{(l)}) \right) \quad (5)$$

The gradient of the error function in Equation 5 is used to adapt the latent representations. The change in the latent representations ($\Delta y_{m,n}^{(l)}$) at a given position $(m, n)$ in layer $l$ is given as

$$\Delta y_{m,n}^{(l)} = \epsilon_{bu} \left( \sum_{i=1,j=1}^{D_{(l-1)}} \ell_p'(y_{(m+i),(n+j)}^{(l-1)} - \hat{y}_{(m+i),(n+j)}^{(l-1)}) \phi'(\mathbf{w}_{m,n,i,j}^{(l)} y_{m,n}^{(l)})(\mathbf{w}_{m,n,i,j}^{(l)})^T \right)$$

$$-\epsilon_{td}(y_{m,n}^{(l)} - \hat{y}_{m,n}^{(l)}) - \epsilon_p \ell_p'(y_{m,n}^{(l)}) \quad (6)$$

where $\ell_p'(.)$ denotes the partial differentiation of the p-norm. $\epsilon_{bu}$ is termed the bottom-up learning rate, $\epsilon_{td}$ is termed the top-down learning rate and $\epsilon_p$ is the learning rate due to regularization. For a given layer $l$, the bottom-up learning rate helps in learning representations that can make better predictions of the representations in the layer below $(l-1)$ and the top-down learning rate helps in learning representations that can be easily predicted by the layer above $(l+1)$. Together, these update terms help in learning sparse latent representations and provide numerical stability to the learning process.

The gradient of the error function in Equation 5 is also used to learn the filter in the network. The change in the filters ($\Delta \mathbf{w}_{m,n,i,j}^{(l)}$) is given as

$$\Delta \mathbf{w}_{m,n,i,j}^{(l)} = \epsilon_w \ell_p'(y_{(m+i),(n+j)}^{(l-1)} - \hat{y}_{(m+i),(n+j)}^{(l-1)}) \phi'(\mathbf{w}_{m,n,i,j}^{(l)} y_{m,n}^{(l)})(y_{m,n}^{(l)})^T - \epsilon_p \ell_p'(\mathbf{w}_{m,n,i,j}^{(l)}) \quad (7)$$

where $\epsilon_w$ is the learning rate.

It may be observed from Equation 6 that the update for latent representations at a given position $(m, n)$ in a particular layer $l$ depends only on the predictions made by the neurons at this position and the filters for this position. Similarly, the update for the filters (Equation 7) associated with location $(m, n)$ depends only on the corresponding latent representations. This allows us to learn the latent representations for the neurons in position $(m, n)$ in layer $l$ and the associated filters in parallel with all the other positions in that layer.

Next, we will describe the update process for the latent representations and the filters using the Equations 6 and 7, respectively. At first the filters are held constant and the latent representations are learned using Equation 6. For a given image and a particular layer, we apply $\kappa$ update steps on the latent representations. This implies that we alternate between computing the error in Equation 5 and updating latent representations (using Equation 6) $\kappa$ times before updating the filters. This approach greatly improves convergence rate of the learning algorithm, reducing the overall number of epochs required for convergence. After these $\kappa$ update steps, the learned latent representations are held constant and a single update step is used to update the filters (using Equation 7). A summary of the learning process is provided in Algorithm 1.

## 3 EXPERIMENTS

In this section, we study the capabilities of the network in inferring the latent representations for a given input image. Firstly, we will study the capabilities of the generative model in reconstructing

**Algorithm 1**

```
 1: while not converged do
 2:     for each input image do
 3:         for each layer (except the input layer) do
 4:             for step = 1 : κ do
 5:                 Update latent representations y_l
 6:             end for
 7:             Update filters w^(l)_{m,n,i,j}
 8:         end for
 9:     end for
10: end while
```

the original images from the inferred latent representations. Secondly, we study the capability of the model to infer the latent representations for an image that is a translated version of the original image. Finally, we analyze the model's abilities in estimating the latent representations for a new image that was not used in training. For this purpose, we trained a 5-layered neural network on 1000 images of horses and ships from the CIFAR-10 data set. The details of the training procedure are provided in Appendix.

## 3.1 GENERATIVE MODEL

The learning algorithm estimates the latent representations for the input images presented to the network at each layer in the model. For a given layer $l$, these latent representations are presented as the output of the neurons in that layer. Based on this, the information is propagated in the network from layer $l$ to the input layer in the network. The output of the input layer neurons produces a reconstruction of the original image. This procedure was repeated for the latent representations estimated at each layer in the model and a reconstruction of the original image was obtained. Figure 5 presents some examples of the images reconstructed using the latent representations at each layer in the trained generative model.

It may be seen from Figure 5 that it is possible to reconstruct the original images using the latent representations generated by the model. However, the images reconstructed by the model are blurry in comparison to the original images. This is a known problem with the mean square error Ledig

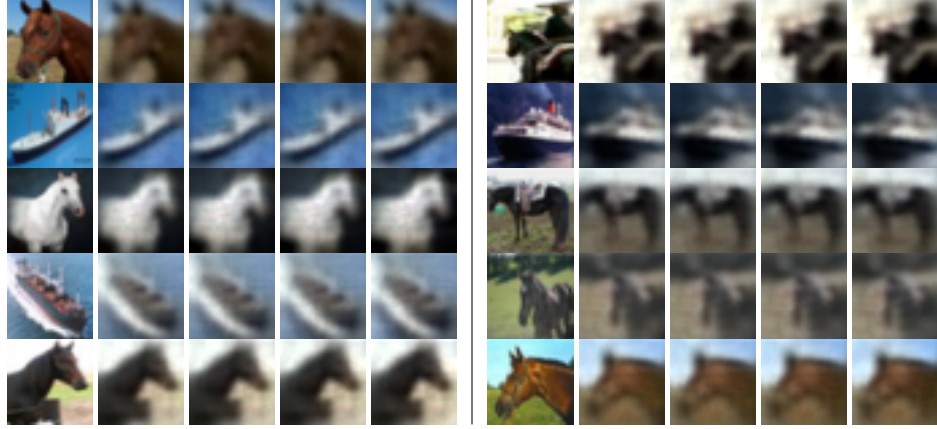

Figure 2: Example images reconstructed by the generative model when the estimated latent representations are presented at corresponding layers in the network. The images are arranged in a table with 2 columns, separated by the vertical bar. Each cell in the table contains a set of five images. The first image in each cell represents the original image from the CIFAR data set and the following 4 images represent the images reconstructed by the model using latent representations generated at layers 1, 2, 3, 4 in the network.

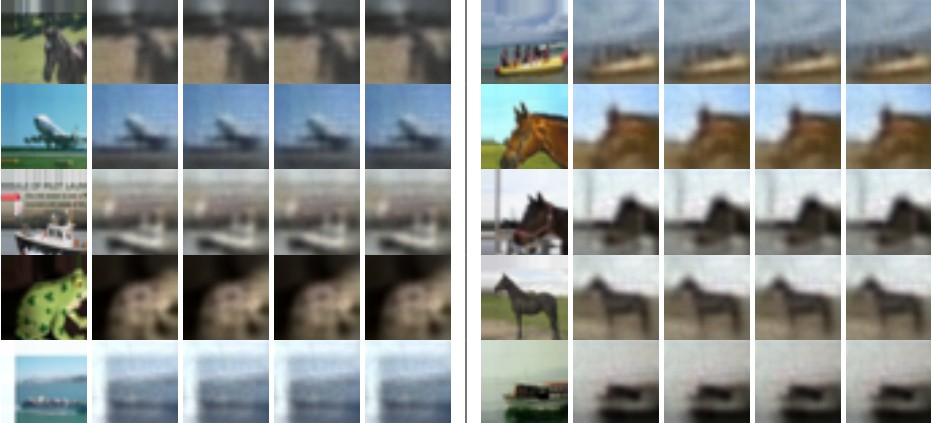

Figure 3: Images reconstructed by the generative model using the latent representations for the translated images. Again, the images are arranged in a table with 2 columns and each cell in the table contains a set of five images. The first image in each cell shows a translated version of the original image from the CIFAR data set and the following 4 images represent the images reconstructed by the model using latent representations generated at different layers in the network.

et al. (2016). It may be possible to obtain visually better images using l1-norm, as suggested in Mathieu et al. (2015).

## 3.2 TRANSLATION INVARIANCE

To study translation invariance in the model, the pixels in the images are shifted to the right and down by 4 pixels. The boundary pixels on the left and top of the original images are used in place of the pixels introduced as a result of shifting the image. For this study, we used images of horses and ships that are used for training as well as images of other objects that are never used in training. These translated images are then presented to the trained generative model and the latent representations for these images are inferred using the Equation 6. Note that in this case the filters in the model are not re-learned. The latent representations for the translated images at each layer in the network are then used to reconstruct the translated images using the procedure described in Section 3.1. Figure 6 shows some examples of the reconstructed images obtained using the latent representations for the translated images.

It can be observed from Figure 6 that the network can generate latent representations for translated images that capture the information present in the input stimulus.

## 3.3 GENERALIZATION

To study generalization, we used the network to infer latent representations of images from the CIFAR-10 data set outside the 1000 images that were used in training. These images are presented to the trained model and the latent representations for these images are inferred using Equation 6. Similar to the previous section, the estimated latent representations at each layer in the network are used to reconstruct the original images using the mechanism described in Section 3.1. Figure 7 presents examples of the images reconstructed from the latent representations that are determined using predictive coding.

It can be seen from Figure 7 that the model can also infer latent representations for images that were never used in training. Furthermore, the generalization ability of the model is not limited to only those objects that are used in training the model. The model can also infer latent representations for objects that are not used training like frog, cars, truck, sparrow, etc (Figure 7). This is due to the retinotopic arrangement of the receptive fields in the network. Such an architecture allows the model to capture granular regularities in real-world images in the lower layers in the network. These granular regularities are common across all real-world images and help the model in generating latent representations for unforeseen objects. Successive layers in the network build upon these regular-

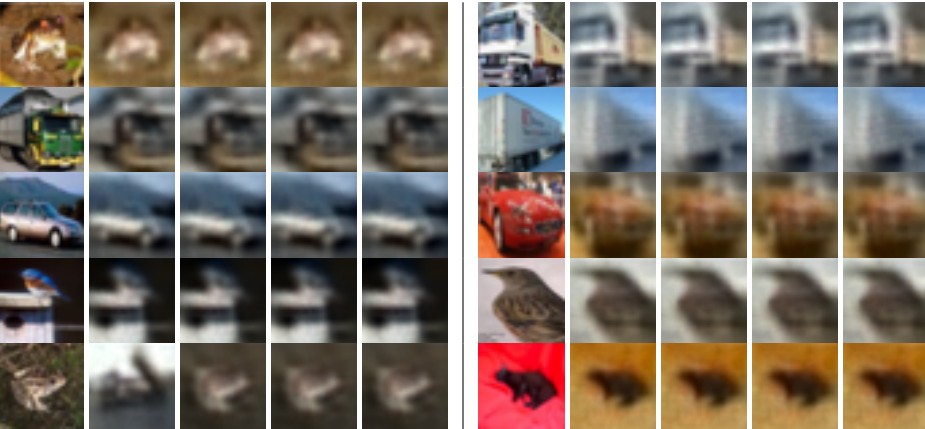

Figure 4: Non-training images reconstructed by the generative model using the latent representations estimated by predictive coding. These images are also arranged in 2 columns and each cell shows the original image and the images reconstructed from its latent representations.

ities to generate more abstract representations of the input images. Note that these generalization properties of the model are achieved while training only on 1000 images whereas most machine learning algorithms rely on large amount of data to improve generalization properties in the model.

## 4 DISCUSSION

In this section, we discuss the computational implications of the algorithm presented in this paper and the similarities it shares with existing approaches in machine learning.

Deep neural networks have improved the state-of-the-art performances in many problems related to image processing like classification, semantic segmentation, etc. These improvements have been achieved by exploiting the availability of cheap computational power. However, with increases in the complexity of neural network architectures, the problem of developing efficient learning algorithms has become prominent. A large body of work in machine learning literature has been dedicated to improving the speed of error-backpropagation which is one of the most used learning algorithms for training deep neural networks. However, an inherent property of error-backpropagation is to systematically propagate information through the network in the forward direction and during learning, propagate the error gradients in the backward direction. This imposes restrictions on the extent of parallelization that can be achieved with error back-propagation.

In this respect, the proposed learning algorithm can be extensively parallelized. It can be observed from Equations 6 and 7 that the latent representations for the neurons in a given layer depend only on the error in predicting the latent representations at the layer below. This aspect of the learning algorithm can be leveraged to update the latent representations and filters at each layer in the network in parallel. Thus the feedforward and feedback processes can be performed at each layer in parallel. Further, the use of a network architecture with retinotopical arrangement of receptive fields allows us to update the latent representations and filters associated with all positions in a given layer in parallel. Thus, the learning algorithm proposed in this paper is amenable to parallelization and can be useful for speeding up the training of deep neural architectures.

Another interesting aspect of the predictive coding is its proximity to the idea of deconvolutional neural networks (Zeiler et al., 2010). Deconvolutional neural networks have also been used to learn the latent representations for a given input image and have been used for the problem of semantic segmentation (Noh et al., 2015). The problem of learning latent representations is inherently an ill-posed problem as there is no unique solution for a given input stimulus. To overcome this issue deconvolutional neural networks optimize on auxiliary variables and the generated latent representations in alternation. A continuation parameter $\beta$ is continuously increased during the learning process until the latent representations are strongly clamped to the auxiliary variables. This requires carefully controlling the learning process and increases the computational requirements of

the learning algorithm due to an extra optimization step on auxiliary variables. Predictive coding provides an alternate solution to this problem. In Equation 6, the update term associated with $\epsilon_{td}$ constraint the learning algorithm to generate latent representations that can be easily predicted by the successive layers in the network. The effect of this constraint on the learning algorithm is same as that of the auxiliary variables in deconvolutional neural networks and impart numerical stability to the learning process. This approach provides a more simpler solution to the problem of learning latent representations without imposing the additional computational effort of optimizing auxiliary variables.

## 5 CONCLUSION

In this paper, we describe a method to train deep neural networks using predictive coding for modeling information processing along cortical sensory hierarchies. The approach uses a neural network in which neurons project only to neurons in their respective receptive fields. This kind of architecture respects the retinotopic arrangement of receptive fields observed in the visual cortical areas.

The method can be used to build a deep generative model for data in any modality. For illustration, we trained the model on a set of real-world images and then used the trained model to infer hierarchical latent representations. Even though the model is trained on a small data set of 1000 images of horses and ships, it can infer effective latent representations for images of other objects like sparrow, cats, trucks, cars, etc. This shows that the trained model is able to capture the statistical regularities present in the real-world images. In this regards, the generalization ability of the model is better than most existing algorithms that usually rely on large amount of data to achieve better generalization.

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

# 6 APPENDIX

## 6.1 TRAINING PROCEDURE

We trained a 5-layered neural network on 1000 images of horses and ships from the CIFAR-10 data set. Each layer in the network used filters of dimension $5 \times 5$. The neurons in a given position $(m, n)$ in layer 1 project to the neurons in input layer through 32 filters ($K_1$). The number of filters in each layer are doubled in each successive layer resulting in 256 filters for the neurons in the last layer. The neurons in all the layers of the network use a linear activation function.

Next, we will describe the procedure for setting the values of different training parameters. At the beginning of training the latent representations at each level in the hierarchy are initialized randomly and there is no causal structure in these latent representations. As a result, the prediction errors are high even for adjacent neurons in a given layer. This results in large update steps which causes problems in the learning process. To avoid these problems, we use a small learning process at the beginning of training and increase it gradually during training. $\epsilon_{bu}$ and $\epsilon_{td}$ are set to 0.0001 at the beginning of training and are increased by a factor of 10 after every 30 epochs until a maximum value of 0.01. $\epsilon_p$ is set to 0.00001 at the beginning of training and is similarly increased by a factor of 10 after every 30 epochs until a maximum value of 0.001. $\epsilon_w$ is set to 0.01 throughout the training process. The losses are always computed using the l2-norm which leads to the blurry reconstructions. Using l1-norm may result in more visually appealing images.

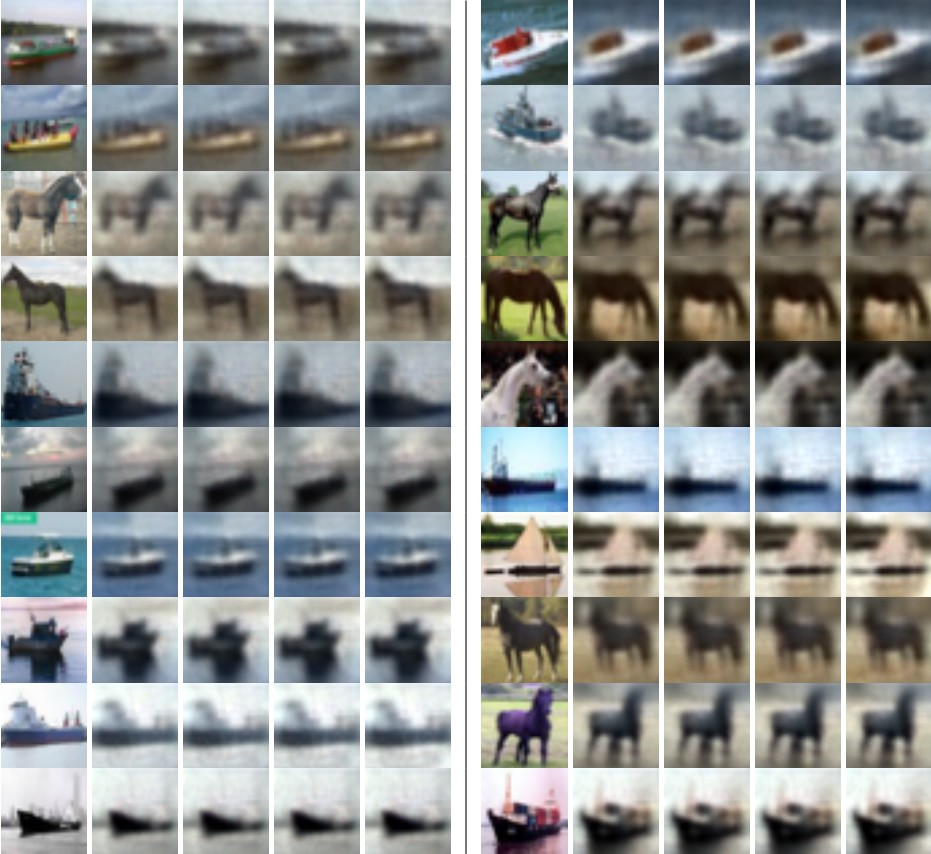

Figure 5: Example images reconstructed by the generative model when the estimated latent causes are presented at corresponding layers in the network. The images are arranged in a table with 2 columns, separated by the vertical bar. Each cell in the table contains a set of five images. The first image in each cell represents the original image from the CIFAR data set and the following 4 images represent the images reconstructed by the model using latent representations generated at layers 1, 2, 3, 4 in the network.

## 6.2 GENERATIVE MODEL

This section presents some more results on generative modeling using the trained model.

## 6.3 TRANSLATION INVARIANCE

This section presents some more results on translation invariance using the trained model.

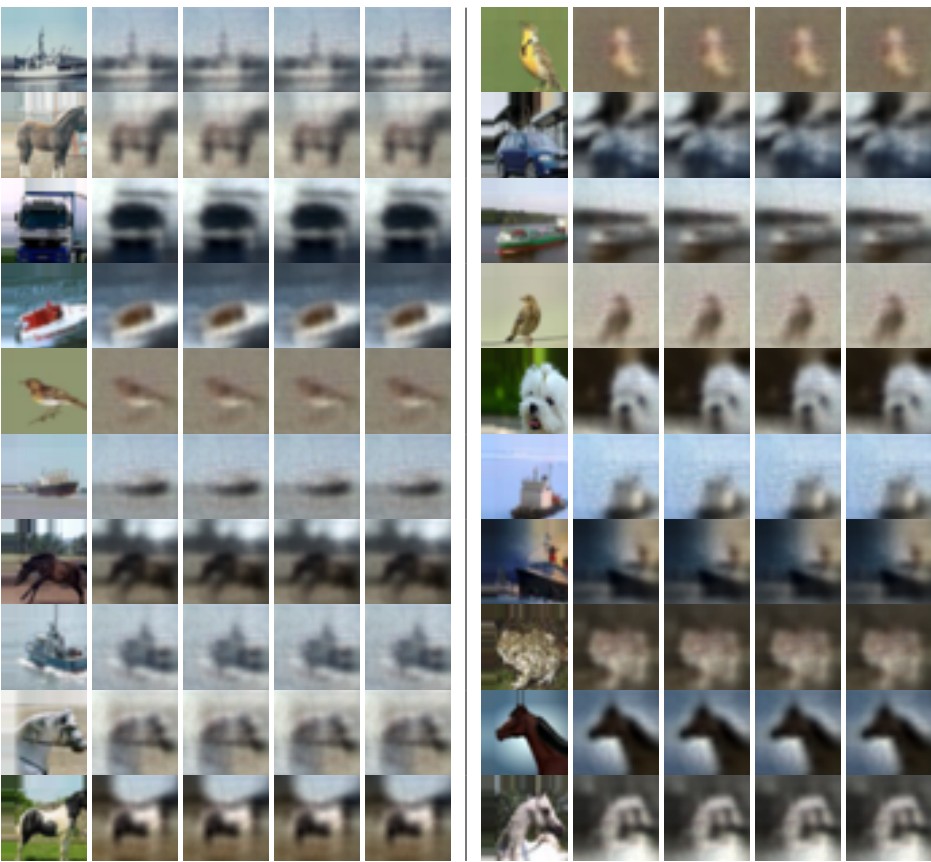

Figure 6: Images reconstructed by the generative model using the latent representations for the translated images. Again, the images are arranged in a table with 2 columns and each cell in the table contains a set of five images. The first image in each cell shows a translated version of the original image from the CIFAR data set and the following 4 images represent the images reconstructed by the model using latent representations generated at different layers in the network.

## 6.4 GENERALIZATION

This section presents some more results on generalization using the trained model.

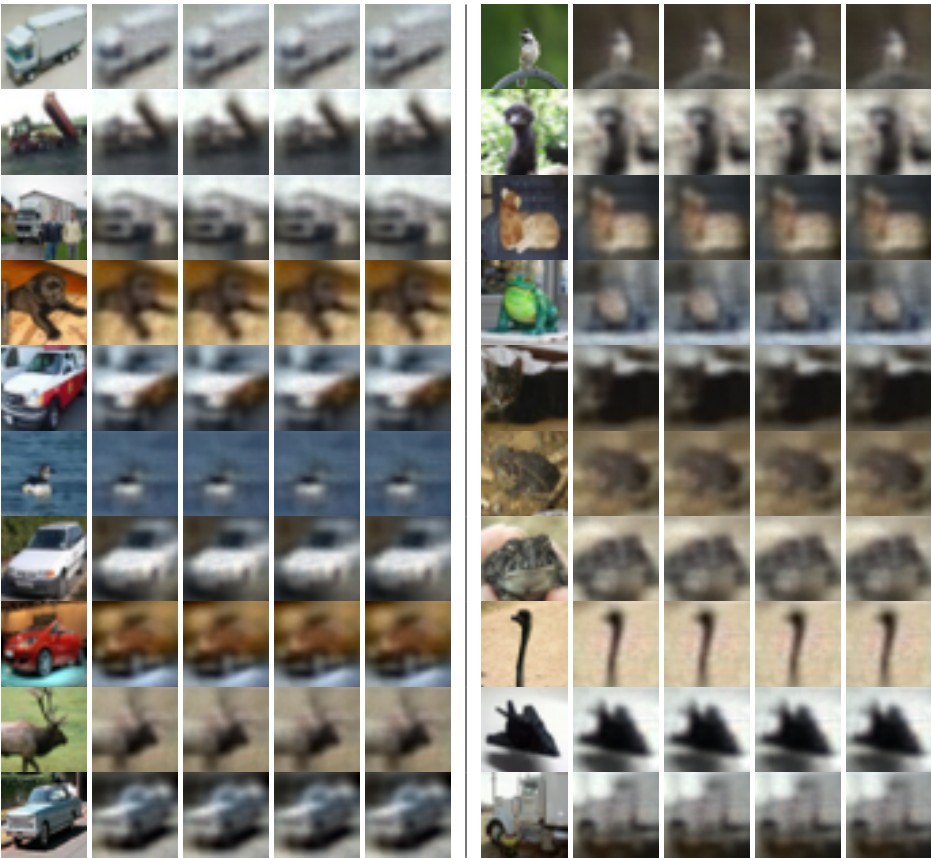

Figure 7: Non-training images reconstructed by the generative model using the latent representations estimated by predictive coding. These images are also arranged in 2 columns and each cell shows the original image and the images reconstructed from its latent representations.

