# OpenReview forum: "A Deep Predictive Coding Network for Learning Latent Representations"
_ICLR.cc/2018/Conference — Reject_

### Official Review · AnonReviewer3 · 2017-11-24
**deep predictive coding**

**Rating:** 4
**Confidence:** 4

**Review:**

Quality

The authors introduce a deep network for predictive coding. It is unclear how the approach improves on the original predictive coding formulation of Rao and Ballard, who also use a hierarchy of transformations. The results seem to indicate that all layers are basically performing the same. No insight is provided about the kinds of filters that are learned.

Clarity

In its present form it is hard to assess if there are benefits to the current formulation compared to already existing formulations. The paper should be checked for typos.

Originality

There exist alternative deep predictive coding models such as https://arxiv.org/abs/1605.08104. This work should be discussed and compared.

Significance

It is hard to see how the present paper improves on classical or alternative (deep) predictive coding results.

Pros

Relevant attempt to develop new predictive coding architectures

Cons

Unclear what is gained compared to existing work.

---

### Official Review · AnonReviewer2 · 2017-11-27
**Major and minor concerns**

**Rating:** 3
**Confidence:** 4

**Review:**

The paper "A Deep Predictive Coding Network for Learning Latent Representations" considers learning of a generative neural network. The network learns unsupervised using a predictive coding setup. A subset of the CIFAR-10 image database (1000 images horses and ships) are used for training. Then images generated using the latent representations inferred on these images, on translated images, and on images of other objects are shown. It is then claimed that the generated images show that the network has learned good latent representations.

I have some concerns about the paper, maybe most notably about the experimental result and the conclusions drawn from them. The numerical experiments are motivated as a way to "understand the capacity of the network with regards to modeling the external environment" (abstract). And it is concluded in the final three sentences of the paper that the presented network "can infer effective latent representations for images of other objects" (i.e., of objects that have not been used for training); and further, that "in this regards, the network is better than most existing algorithms [...]".

I expected the numerical experiments to show results instructive about what representations or what abstractions are learned in the different layers of the network using the learning algorithm and objectives suggested. Also some at least quantifiable (if not benchmarked) outcomes should have been presented given the rather strong claims/conclusions in abstract and discussion/conclusion sections. As a matter of fact, all images shown (including those in the appendix) are blurred versions of the original images, except of one single image: Fig. 4 last row, 2nd image (and that is not commented on). If these are the generated images, then some reconstruction is done by the network, fine, but also not unsurprising as the network was told to do so by the used objective function. What precisely do we learn here? I would have expected the presentation of experimental results to facilitate the development of an understanding of the computations going on in the trained network. How can the reader conclude any functioning from these images? Using the right objective function, reconstructions can also be obtained using random (not learned) generative fields and relatively basic models. The fact that image reconstruction for shifted images or new images is evidence for a sophisticated latent representations is, to my mind, not at all shown here. What would be a good measure for an "effective latent representation" that substantiates the claims made? The reconstruction of unseen images is claimed central but as far as I could see, Figures 2, 3, and 4 are not even referred to in the text, nor is there any objective measure discussed. Studying the relation between predictive coding and deep learning makes sense, but I do not come to the same (strong) conclusions as the author(s) by considering the experimental results - and I do not see evidence for a sophisticated latent representation learned by the network. I am not saying that there is none, but I do not see how the presented experimental results show evidence for this.

Furthermore, the authors stress that a main distinguishing feature of their approach (top of page 3) is that in their network information flows from latent space to observed space (e.g. in contrast to CNNs). That is a true statement but also one which is true for basically all generative models, e.g., of standard directed graphical models such as wake-sleep approaches (Hinton et al., 1995), deep SBNs and more recent generative models used in GANs (Goodfellow et al, 2014). Any of these references would have made a lot of sense.

With my evaluation I do not want to be discouraging about the general approach. But I can not at all give a good evaluation given the current experimental results (unless substantial new evidence which make me evaluate these results differently is provided in a discussion).


Minor:

- no legend for Fig. 1

-notes -> noted

have focused

---

### Official Review · AnonReviewer1 · 2017-11-28
**Good overall idea but needs further development**

**Rating:** 3
**Confidence:** 5

**Review:**

The paper attempts to extend the predictive coding model to a multilayer network.  The math is developed for a learning rule, and some demonstrations are shown for reconstructions of CIFAR-10 images.

The overall idea and approach being pursued here is a good one, but the model needs further development.  It could also use better theoretical motivation - i.e., what sorts of representations do you expect to emerge in higher layers?  Can you demonstrate this with a toy example and then extend to real data?

That the model can reconstruct images per se is not particularly interesting.  What we would like to see is that it has somehow learned a more useful or meaningful representation of the data.  For example, what do the learned weights look like?  That would tell you something about what has been learned.

---

### Decision · Program_Chairs · 2018-01-29
**ICLR 2018 Conference Acceptance Decision**

**Decision:**

Reject

**Comment:**

The paper attempts to develop a method for learning latent representations using deep predictive coding and deconvolutional networks. However, the theoretical motivation for the proposed model in relation to existing methods (such as original predictive coding, deconvolutional networks, ladder networks, etc.), as well as the empirical comparison against them is unclear. The experimental results on the CIFAR10 dataset do not provide much insight on what kind of meaningful/improved representations can be learned in comparison to existing methods, both qualitatively and quantitatively. No rebuttal was provided.